# National Early Warning Score 2 (NEWS2) to predict poor outcome in hospitalised COVID-19 patients in India

Pugazhvannan CR[1], Ilavarasi Vanidassane[2], Dhivya Pownraj[1], Ravichandran Kandasamy[3], Aneesh Basheer[1]*

1 Department of General Medicine, Pondicherry Institute of Medical Sciences, Puducherry, India,
2 Department of Medical Oncology, Pondicherry Institute of Medical Sciences, Puducherry, India,
3 Department of Biostatistics, Pondicherry Institute of Medical Sciences, Puducherry, India

* basheeraneesh@gmail.com

**Data Availability Statement:** All relevant data are within the manuscript and its supporting information files.

## Abstract

### Background

While several parameters have emerged as predictors of prognosis of COVID-19, a simple clinical score at baseline might help early risk stratification. We determined the ability of National Early Warning Score 2 (NEWS2) to predict poor outcomes among adults with COVID-19.

### Methods

A prospective study was conducted on 399 hospitalised adults with confirmed SARS-CoV-2 infection between August and December 2020. Baseline NEWS2 score was determined. Primary outcome was poor outcomes defined as need for mechanical ventilation or death within 28 days. The sensitivity, specificity and Area under the curve were determined for NEWS2 scores of 5 and 6.

### Results

Mean age of patients was 55.5 ± 14.8 years and 275 of 399 (68.9%) were male. Overall mortality was 3.8% and 7.5% had poor outcomes. Median (interquartile range) NEWS2 score at admission was 2 (0–6). Sensitivity and specificity of NEWS 2 of 5 or more in predicting poor outcomes was 93.3% (95% CI: 76.5–98.8) and 70.7% (95% CI: 65.7–75.3) respectively [area under curve 0.88 (95% CI: 0.847–0.927)]. Age, baseline pulse rate, baseline oxygen saturation, need for supplemental oxygen and ARDS on chest X ray were independently associated with poor outcomes.

### Conclusions

NEWS2 score of 5 or more at admission predicts poor outcomes in patients with COVID-19 with good sensitivity and can easily be applied for risk stratification at baseline. Further studies are needed in the Indian setting to validate this simple score and recommend widespread use.

**Funding:** The authors received no specific funding for this work.

**Competing interests:** The authors have declared that no competing interests exist.

## Introduction

More than a year has passed since COVID-19 was declared a pandemic. While majority of SARS-CoV-2 infections are asymptomatic or mild, a small proportion develop severe disease that is often fatal despite best available care [1]. Mortality in this group of patients is high [2]. It is estimated that around 15–20% of hospitalised patients with COVID 19 need ICU care or die of severe disease [1,3]. Increasing age and co-morbidities such as diabetes, obesity, cardiovascular and lung disease have been linked to severe disease and death in many studies [4].

Presence of risk factors and clinical presentation is currently being used to triage patients at the time of admission into mild, moderate and severe disease. Further several laboratory parameters like the C-reactive protein, Ferritin and D-dimer have emerged as potential markers of severity [1,5–7], although none have a definite prognostic value. In other diseases such as sepsis and critical illnesses, several scoring systems have been validated for predicting poor outcomes and mortality. The National Early Waring Score 2 (NEWS2) is one such tool that is simple and enables health care staff to identify high risk patients and escalate care [8]. Some bodies have suggested the use of NEWS2 along with clinical examination to triage COVID-19 patients albeit with caution [9]. Such risk stratification may help quicker decision making and enable the treating doctors to divert more attention, time and resources to those identified as high risk for fatal outcome. Unlike most other scores, NEWS2 also includes oxygenation criteria such as hypoxia and supplemental oxygen requirement, which are particularly important in assessment of COVID-19 patients. All these make NEWS2 a good baseline indicator to be explored as a potential predictor of severe disease and death among COVID-19 patients. We therefore aimed to determine whether NEWS 2 score at admission predicts poor outcome in patients with COVID-19 disease.

## Methods

A prospective cohort study was conducted between August and December 2020 at the Pondicherry Institute of Medical Sciences, a tertiary care teaching hospital also functioning as a designated testing and treating centre for COVID-19. Inclusion criteria was adults admitted with a diagnosis of COVID-19 confirmed by detection of SARS-CoV-2 by RT-PCR. Following written informed consent, consecutive eligible participants were interviewed to obtain demographic details such as age, residence and contact with confirmed or suspected COVID-19 cases in household or workplace. Presenting symptoms with duration, associated co-morbidities and treatment for the same were documented. At admission, the vital signs including blood pressure, pulse, respiratory rate and oxygen saturation were recorded. The NEWS2 score was calculated on the following parameters: respiratory rate, oxygen saturation (SpO2), need for supplemental oxygen, pulse rate, level of consciousness and temperature (Fig 1). We also collected baseline laboratory investigations as part of routine COVID 19 care. These patients were followed up on a daily basis for improvement/deterioration. We classified patient outcomes at day 28 as discharged, hospitalised but without oxygen, hospitalised on supplemental oxygen, hospitalised on High flow nasal oxygen (HFNO) or Non-invasive ventilation (NIV), mechanically ventilated and expired. The primary outcome was the ability of NEWS2 score at admission to predict poor outcome in patients with COVID 19 disease, defined as need for mechanical ventilation or death within 28 days. Secondary outcomes were association of other clinical (such as diabetes, hypertension, age and chronic lung, kidney or liver diseases) and laboratory variables with poor outcomes.

Data were collected using the Epicollect 5 application on tablets in order to minimise contact with contaminated documents. We described the frequency of clinical and laboratory features using means and standard deviations, and proportions with confidence intervals. The sensitivity,

**Fig 1. NEWS2 scoring matrix.** Reproduced from: Royal College of Physicians. *National Early Warning Score (NEWS) 2*: *Standardising the assessment of acute-illness severity in the NHS*. Updated report of a working party. London: RCP, 2017. The final score is a composite of the points for individual criteria. Higher scores (of 5 and above) usually indicate need for escalating care.

specificity, positive and negative predictive value with 95% confidence intervals (CIs) of NEWS2 score at admission in predicting poor outcomes were calculated. We used cut off scores of 5 and 6 for the NEWS2 score for these calculations. Hence for calculating sensitivity at a cut off score of 5, we determined the proportion of patients with NEWS2 score of 5 or more who died or needed mechanical ventilation. Similarly for a cut off of 6 and above we determined the proportion of patients with an admission score of 6 and above who died or needed mechanical ventilation. For calculation of specificities, we determined proportions of patients with these scores who did not have a poor outcome. Univariate analysis was performed with Chi-square test/Fisher's Exact Test to determine associations between clinical and laboratory parameters and poor outcome of categorical variables. Mann-Whitney test/ t test was done for continuous variables based on normality condition. Independent associations between potential variables and poor outcome were determined and expressed as Odds ratios with 95% Confidence intervals and p values. In this model, those who needed mechanical ventilation or died within 28 days (i.e., poor outcome) were considered as case-patients and patients discharged (i.e., good outcome) were considered as control-patients. Exposure status was defined as potential variables known to affect outcomes such as comorbidities, baseline clinical parameters, laboratory parameters and findings of chest X-ray. Logistic regression was employed to quantify the relationship between these variables and poor outcomes and expressed as Odds ratio with 95% Confidence intervals and p values. Variables which were significant at $p < 0.1$ in univariate analysis were considered for multiple logistic regression analysis. p value $< 0.05$ was used to define statistical significance.

The study was approved by the PIMS Institute Ethics Committee (IEC:RC/2020/72) and all data pertaining to this study are available within this article and supplementary information file (S1 Dataset).

## Results

During the period between August 2020 and December 2020, 416 patients were admitted with a positive RT-PCR test for SARS-CoV-2. Among these, 16 did not consent for the study and one was referred elsewhere; finally, 399 eligible patients were included in the analysis.

The mean age of patients was 55.5 ± 14.8 years and 199 of 399 (49.9%) were between 40 to 60 years of age. 143 of 399 (35.8%) were over 60 years of age. 275 of 399 (68.9%) patients were males. The most common symptom was fever [304/399 (76.2%)] followed by cough (Table 1). 199 of 399 (49.9%) patients were diabetics while 176 of 399 (44.1%) had hypertension. Table 2 presents the common comorbidities of patients.

94 out of 399 (23.6%) patients had a history of contact with confirmed case of COVID-19. Based on the WHO clinical criteria, 53 out of 399 (13.3%) patients had severe disease. The overall 28-day mortality was 3.8% (15 out of 399 patients). Other outcomes of patients at day 28 is summarised in Table 3. Poor outcome (defined as death or mechanical ventilation at any time during 28 days) occurred in 30 of the 399 (7.5%) patients. Among these, death occurred in 15 patients. The remaining 15 required mechanical ventilation at some point of time during hospitalisation. This includes 2 patients who needed mechanical ventilation but could be weaned off before day 28 and 13 patients who were still on invasive ventilation at day 28.

The mean admission NEWS2 score was 3.3 ± 3.3 and median (IQR) NEWS2 score at admission was 2 (0–6). The baseline NEWS2 score was 5 or more in 136 of the 399 (34.1%) patients while it was 6 or more in 115 of the 399 (28.8%) patients.

## Sensitivity and specificity of NEWS2 score

We determined predictive accuracy of NEWS2 at score of 5 as well as 6. Accordingly, 28 out of 30 patients with poor outcomes had a NEWS2 score of 5 or more while only 2 had score less than 5. Similarly, a NEWS2 score of 5 or more was seen in 108 out of 369 patients with good outcome while 261 of them had a score of less than 5. Thus the sensitivity, specificity, positive predictive value and negative predictive value of an admission NEWS2 score of 5 or more to predict poor outcome was 93.3% (95% CI: 76.5–98.8), 70.7% (95% CI: 65.7–75.3), 20.6% (95% CI: 14.3–28.5) and 99.2% (955 CI: 97.0–99.9) respectively (S1 Table). Using a score of 6 as cut off, 27 out of 30 patients with poor outcomes had NEWS2 score of 6 or more compared to 3 who had a score less than 6. NEWS2 score of 6 or more had a sensitivity of 90% (95% CI: 72.3–97.4) and specificity of 76.2% (95% CI: 71.4–80.3) (Table 4). The Area under the curve (AUC) to predict poor outcome was also high (0.887; 95% CI: 0.847–0.927) which is considered excellent for discrimination (Fig 2).

On univariate analysis, age, chronic kidney disease, chronic heart disease, need for supplemental oxygen at admission and ARDS on baseline Chest X ray were associated with poor outcomes. Other factors associated with poor outcome included baseline pulse rate, baseline respiratory rate, baseline oxygen saturation, leucocyte counts, Neutrophil to Lymphocyte ratio, serum creatinine, ESR, CRP, D-Dimer, Ferritin, Blood urea nitrogen and AST (Table 5). However, on multiple logistic regression only the following factors were associated with poor

**Table 1. Frequency of symptoms of patients at presentation (n = 399).**

| Symptom | Number (%) |
|---|---|
| Fever | 304 (76.2) |
| Cough | 229 (57.4) |
| Sore throat | 134 (33.6) |
| Breathlessness | 147 (36.8) |
| Myalgia | 202 (50.6) |
| Diarrhea | 51 (12.8) |
| Loss of taste | 80 (20.1) |
| Loss of smell | 67 (16.8) |

**Table 2. Distribution of co-morbidities among patients admitted with COVID-19 (n = 399).**

| Comorbidity | Number (%) |
|---|---|
| Diabetes | 199 (49.9) |
| Systemic hypertension | 176 (44.1) |
| Chronic heart disease | 37 (9.3) |
| Chronic kidney disease | 17 (4.3) |
| Chronic lung disease | 11 (2.8) |
| Chronic liver disease | 3 (0.8) |
| Others* | 30 (7.5) |

*Others include hypothyroidism, malignancy, pancreatitis, seizure disorder, post-renal transplant and depression.

outcomes: age, baseline pulse rate, baseline oxygen saturation, need for supplemental Oxygen, and ARDS on Chest X ray (Table 6). Eight of 30 (26.7%) patients with poor outcome had ARDS compared to 3 of 369 (0.8%) who had good outcomes, indicating that ARDS on admission was strongly associated with poor outcomes (Odds ratio– 17.2).

## Discussion

This study conducted at a tertiary care hospital that tests and treats COVID-19 disease of all degrees of severity identified that NEWS2 score applied to patients at admission has high sensitivity and reasonable specificity to predict progression to mechanical ventilation or death at 28 days. A score of 5 or more at baseline had maximum area under the curve for prediction of poor outcomes. Further, we found that increasing age, need for supplemental oxygen at baseline, pulse rate at baseline and an initial chest X ray showing features of ARDS were independently associated with poor outcomes.

Since the beginning of COVID-19 pandemic several studies have identified factors associated with risk of death. Early studies from Wuhan indicated that older age, smoking, admission body temperature, Neutrophil to Lymphocyte (NLR) ratio, platelet counts, D-dimer and serum creatinine correlated with increased risk of death from COVID-19 [10]. Another small retrospective study early in the pandemic identified higher Sequential Organ Failure Assessment (SOFA) scores to be a predictor of death in addition to older age and high D-dimer levels [4]. As more data emerged, it was evident that death in COVID-19 disease was due to pulmonary as well as non-pulmonary complications such as acute cardiac injury and heart failure, and data from large cohorts using electronic health records showed that male gender, uncontrolled diabetes and severe asthma were strongly associated with death [11]. Since then a long

**Table 3. Outcomes at day 28 of patients admitted with COVID-19 (n = 399).**

| Outcome | Number (%) |
|---|---|
| Discharged/hospitalized with no supplemental Oxygen | 268 (67.2) |
| Hospitalized with supplemental Oxygen | 77 (19.3) |
| Hospitalized with HFNO* or NIV# | 26 (6.5) |
| Mechanically ventilated | 13 (3.3) |
| Dead | 15 (3.8) |

*HFNO–high flow nasal oxygen.

#NIV–non-invasive ventilation.

**Table 4. Comparison of diagnostic accuracy parameters of scores of 5 and 6.**

| Characteristic | NEWS2 score of 5 or more (136; 34.1%) | NEWS2 score of 6 or more (115; 28.8%) |
|---|---|---|
| Sensitivity | 93.3% (95% CI: 76.5–98.8) | 90.0% (95% CI: 72.3–97.4) |
| Specificity | 70.7% (95% CI: 65.7–75.3) | 76.2% (95% CI: 71.4–80.3) |
| Positive predictive value | 20.6% (95% CI: 14.3–28.5) | 23.5% (95% CI: 16.3–32.5) |
| Negative predictive | 99.2% (95% CI: 97.0–99.9) | 98.9% (95% CI: 96.9–99.6) |

list of factors attributable to mortality in COVID-19 has emerged including several laboratory parameters such as D-dimer [2].

Despite this, it has been a tough task to predict which patients progress to death or poor outcome such as need for mechanical ventilation [12], especially at the time of admission or first encounter. Many studies reported markers such as D-dimer, Lactate Dehydrogenase (LDH) and interleukins to be associated with severe disease and death; however, most of these are either unavailable or prohibitively expensive to be used routinely for risk stratification [13].

While NEWS and NEWS2 scores have been in vogue in the United Kingdom and many other countries for the triaging and monitoring of hospitalised patients, its use in India and many other nations has not been widespread. Initially used in 2012 for identifying and monitoring sick patients in hospital, the NEWS score was modified in 2017 to NEWS2 by including confusion and oxygen saturation [14]. Shortly after the beginning of COVID-19 pandemic, the Royal College of Physicians UK issued guidance advocating use of NEWS2 score for managing patients with COVID-19 [15]. However, this recommendation was not based on existing evidence but an extrapolation of its use in the pre-pandemic era. This led experts to advise caution against its use in COVID-19 setting particularly in primary care [9]. Subsequently, Myrstad and colleagues determined the ability of NEWS2 score at emergency room admission to predict severe disease and death in COVID-19 patients [16]. NEWS2 score of 6 or more had 80% sensitivity and 84.3% specificity in predicting severe disease, with an area under the curve (AUC) of 0.822. But this study was limited by small sample size (66 patients) and retrospective data collection for certain variables such as comorbidities. Similarly, another small

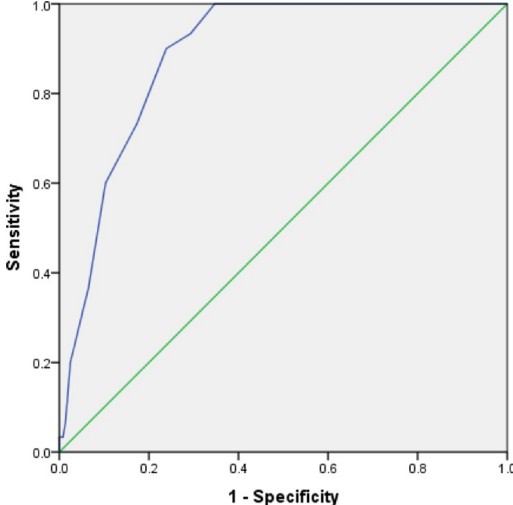

**Fig 2. Receiver Operating Characteristics (ROC) curve showing area under curve for NEWS2 scores in predicting poor outcomes among COVID-19 patients.**

**Table 5. Univariate analysis of factors associated with poor outcomes at day 28.**

| Variables | N (percentage to total) | Good outcome (n = 369) | Poor outcome (n = 30) | p value |
|---|---|---|---|---|
| Age* | - | 54.7 ± 14.8 | 65.3 ± 10.6 | < 0.001 |
| Male@ | 275 (68.9%) | 252 (68.3%) | 23 (76.7%) | 0.341 |
| Diabetes@ | 199 (49.9%) | 180 (48.8%) | 19 (63.3%) | 0.125 |
| Hypertension@ | 176 (44.1%) | 161 (43.6%) | 15 (50.0%) | 0.499 |
| Chronic Lung Disease@ | 11 (2.8%) | 11 (3.0%) | 0 (0%) | 1.000 |
| Chronic Liver Disease@ | 3 (0.8%) | 3 (0.8%) | 0 (0%) | 1.000 |
| Chronic Kidney Disease@ | 17 (4.3%) | 13 (3.5%) | 4 (13.3%) | 0.031 |
| Chronic Heart Disease@ | 37 (9.3%) | 31 (8.4%) | 6 (20.0%) | 0.047 |
| Need for oxygen supplementation@ | 146 (36.6%) | 118 (32.0%) | 28 (93.3%) | < 0.001 |
| Chest X-ray ARDS@ | 11 (2.8%) | 3 (0.8%) | 8 (26.7%) | < 0.001 |
| Baseline Pulse rate | - | 90.0 (81.5–101.0) | 110 (93.8–116.8) | < 0.001 |
| Baseline systolic | - | 130.0 (120.0–140.0) | 125.0 (120.0–140.0) | 0.950 |
| Baseline Diastolic | - | 80.0 (70.0–90.0) | 80.0 (70.0–82.5) | 0.368 |
| Baseline respirate rate | - | 21.0 (18.0–26.0) | 30.0 (25.0–30.0) | < 0.001 |
| Baseline oxygen saturation | - | 96.0 (93.0–98.0) | 87.0 (79.5–93.0) | < 0.001 |
| Haemoglobin | - | 13.0 (11.4–14.3) | 12.5 (11.0–13.6) | 0.131 |
| Total Leukocyte count | - | 5840.0 (4600.0–7650.0) | 8200.0 (4950.0–11812.5) | 0.005 |
| Neutrophil to Lymphocyte ratio | - | 3.0 (2.0–4.0) | 4.0 (3.4–9.1) | < 0.001 |
| ESR | - | 29.0 (11.0–51.5) | 46.0 (28.0–82.0) | < 0.001 |
| CRP | | 12.0 (4.0–36.0) | 30.0 (18.5–96.0) | < 0.001 |
| S Ferritin | | 278.0 (112.0–602.5) | 695.5 (341.8–1161.0) | < 0.001 |
| D Dimer | | 0.47 (0.25–0.87) | 1.17 (0.71–2.70) | < 0.001 |
| AST | | 28.0 (20.0–43.0) | 38.5 (29.5–58.5) | 0.009 |
| ALT | | 27.0 (19.0–41.0) | 32.5 (24.5–41.3) | 0.064 |
| BUN | | 18.0 (12.0–27.0) | 25.0 (20.0–47.5) | 0.001 |
| S Creatinine | | 0.9 (0.7–1.0) | 1.0 (0.8–1.3) | 0.001 |

*: Mean ± standard deviation; @: Number (percentage); remaining median (inter quartile range).

retrospective study from Italy showed NEWS2 score at admission to be a good predictor of ICU admission (AUC of 0.90; 95% CI 0.82–0.97) [17]. An intensive care specialist team from China proposed using a modified NEWS2 score for triaging patients by including age above 65 years since older age stood out as an independent risk factor in most studies [18]. However, this scoring system has not been validated as yet.

Our study in contrast was done on a prospective cohort of 399 patients with COVID-19 and determined NEWS2 score at admission for all of them. We used need for mechanical ventilation or death anytime during 28 days as poor outcome. Further, despite being a hospital-based study, our cohort included patients with varying severity of COVID-19, including mild cases since government guidelines during the study period enabled admission of relatively stable patients as well. We found an admission NEWS 2 score of 5 and above to be better than 6 in contrast to Myrstad et al. At this cut-off, AUC for predicting poor outcomes was high (0.887; 95% CI: 0.847–0.927) and higher than the AUC for score of 6 found by Myrstad et al [16].

Carr and others studied modified versions of NEWS2 score by adding age and a set of other routine blood tests at admission to discriminate severe COVID-19 disease at 14 days [19]. While the former model had poor-to-moderate discrimination, the latter model was affected by calibration issues at different study sites. Adding pre-existing co-morbidities to the model

**Table 6. Multiple logistic regression of factors associated with poor outcome at day 28.**

| Variable | Odds ratio | 95% CI | p value |
| --- | --- | --- | --- |
| Age | 1.048 | 1.003–1.095 | 0.036* |
| Chronic heart disease | 1.007 | 0.247–4.107 | 0.992 |
| Chronic kidney disease | 0.663 | 0.087–5.066 | 0.692 |
| Need for oxygen supplementation | 6.071 | 1.114–33.070 | 0.037* |
| Baseline pulse | 1.040 | 1.003–1.078 | 0.034* |
| Baseline respiratory rate | 0.988 | 0.932–1.048 | 0.692 |
| Baseline oxygen saturation | 0.933 | 0.873–0.997 | 0.039* |
| Total leucocyte count | 1.00 | 1.00–1.00 | 0.085 |
| Neutrophil to Lymphocyte ratio | 1.073 | 0.990–1.163 | 0.088 |
| ESR | 1.006 | 0.989–1.024 | 0.464 |
| CRP | 1.004 | 0.993–1.015 | 0.461 |
| Ferritin | 1.00 | 0.999–1.001 | 0.566 |
| D-Dimer | 1.034 | 0.933–1.144 | 0.526 |
| Blood urea nitrogen | 1.003 | 0.986–1.019 | 0.743 |
| AST | 0.998 | 0.969–1.027 | 0.869 |
| ALT | 1.002 | 0.975–1.030 | 0.868 |
| ARDS on Chest X ray | 17.238 | 2.186–135.929 | 0.007* |

*Denotes statistically significant variables (p < 0.05).

Note: Variables with p value less than or equal to 0.1 on univariate analysis were chosen for multiple logistic regression.

made no difference to risk prediction either. This suggests that adding age or co-morbidities to the NEWS2 score adds little to improve its value, justifying our decision to test the standard score. Further, we found it useful for a longer-term outcome of 28 days.

A retrospective study on 296 hospitalised adults with COVID-19 from a single centre in UK found that NEWS2 score of 5 or more anytime during stay predicted the occurrence of deterioration with a sensitivity and specificity of 0.98 (95% CI 0.96–1.00) and 0.28 (95% CI 0.21–0.35) respectively [20], emphasising its utility in longitudinal monitoring of COVID-19 patients as well. However, the caveat is a high false alarm rate. A larger study that evaluated performance of NEWS and NEWS2 scores among five admission cohorts also demonstrated good discrimination for death or ICU admission within 24 hours for patients with COVID-19 [21]. These results also suggest that NEWS2 score may be used without any modifications in COVID-19 settings.

Marta et al determined the value of NEWS 2 score at admission among 477 in-patients with COVID-19 in predicting in-hospital mortality. Their findings were similar to this study with a score more than 5 being the best cut-off [AUC—0.84 (95% CI 0.79–0.90)] [22]. The in-hospital mortality was higher (11.5%) than our study. On the other hand, ROX index, a simple score based on oxygenation and respiratory rate outperformed NEWS2 score in terms of predicting deterioration in COVID-19 patients [23]. These results however need to be interpreted with caution in view of retrospective design and use of a convenience sample. Another recent study that compared index NEWS and NEWS2 scores found low discrimination for COVID-19 versus non-COVID-19 patients; however, there was higher risk of mortality for COVID-19 patients than non-COVID-19 patients for each value of the admission NEWS2 score [24].

In this study, we also determined associations between potential risk factors and poor outcomes. Univariate analysis yielded 18 variables with possible significant association with risk of mechanical ventilation or death. However, after multivariate analysis only age above 60

years, baseline pulse, oxygen saturation, need for supplemental oxygen and admission chest X ray evidence of ARDS remained significantly associated. Several laboratory parameters found in other studies as poor prognostic indicators including D-dimer, serum Ferritin and CRP were not associated with risk of poor outcome in this study. This could be related to the heterogeneity in outcomes chosen in different studies as well as the high prevalence of co-morbidities like diabetes and hypertension in our population in general with rise in metabolic syndrome and lifestyle diseases.

External validity of our results may be limited by the single centre nature of this study. Further some of the co-morbidities were determined based on history such as diabetes since variables like glycated haemoglobin were not available for all patients. However, the large sample size and the inclusion of patients with all grades of severity of COVID-19 increases generalisability. Moreover, to the best of our knowledge, this is the first large study from India on the usefulness of NEWS2 score in COVID-19 patients. Being a simple scoring system based on physiological variables that can easily be determined even in remote healthcare settings, it could help doctors identify patients at risk of worsening.

## Conclusion

NEWS2 score of 5 or more at admission predicts mortality and need for mechanical ventilation in COVID-19 patients with a high sensitivity of 93.3% and reasonable specificity of 70.7%. ARDS at admission is strongly associated with risk of poor outcomes; older age, low baseline oxygen saturation and need for supplemental oxygen being other risk factors. Patients with these features must be monitored intensively to detect worsening earlier and institute evidence-based measures available. This score must be further validated in Indian settings on a larger scale and put to use especially in resource poor settings to identify patients at need for referral to tertiary care centres or transfer to intensive care units.

## Supporting information

**S1 Table. Sensitivity and specificity of NEWS2 scores of 5 and 6 in predicting poor outcomes (death or need for mechanical ventilation) among COVID-19 patients.**
(DOCX)

**S1 Dataset. Raw data sheet of the study with identifying information removed.**
(XLSX)

## Acknowledgments

We gratefully acknowledge the assistance from the nurses and support staff working in the COVID wards in collection of data and follow up of patients. We thank Dr. Kurien Thomas, Professor and Head of Medicine, Pondicherry Institute of Medical Sciences for his guidance in formulating the research question and developing the protocol.

## Author Contributions

**Conceptualization:** Aneesh Basheer.

**Data curation:** Pugazhvannan CR, Ilavarasi Vanidassane, Dhivya Pownraj.

**Formal analysis:** Ravichandran Kandasamy.

**Investigation:** Pugazhvannan CR, Ilavarasi Vanidassane, Dhivya Pownraj, Ravichandran Kandasamy, Aneesh Basheer.

**Methodology:** Ilavarasi Vanidassane, Dhivya Pownraj, Ravichandran Kandasamy, Aneesh Basheer.

**Project administration:** Ilavarasi Vanidassane, Dhivya Pownraj, Ravichandran Kandasamy, Aneesh Basheer.

**Software:** Pugazhvannan CR, Ilavarasi Vanidassane, Ravichandran Kandasamy.

**Supervision:** Aneesh Basheer.

**Writing – original draft:** Aneesh Basheer.

**Writing – review & editing:** Pugazhvannan CR, Dhivya Pownraj, Ravichandran Kandasamy.

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
