## [Decision Letter · Decision Letter 0]

26 Aug 2021

PONE-D-21-23108

National Early Warning Score 2 (NEWS2) to predict poor outcome in hospitalised COVID-19 patients in India

PLOS ONE

Dear Dr. Basheer,

Thank you for submitting your manuscript to PLOS ONE. After careful consideration, we feel that it has merit but does not fully meet PLOS ONE’s publication criteria as it currently stands. Therefore, we invite you to submit a revised version of the manuscript that addresses the points raised during the review process.

Please revise accordingly.

We look forward to receiving your revised manuscript.

Kind regards,

Academic Editor

PLOS ONE

Journal Requirements:

Reviewers' comments:

Reviewer's Responses to Questions

**Comments to the Author**

1. Is the manuscript technically sound, and do the data support the conclusions?

Reviewer #1: Partly

Reviewer #2: Yes

2. Has the statistical analysis been performed appropriately and rigorously? 

Reviewer #1: I Don't Know

Reviewer #2: Yes

3. Have the authors made all data underlying the findings in their manuscript fully available?

Reviewer #1: No

Reviewer #2: Yes

4. Is the manuscript presented in an intelligible fashion and written in standard English?

Reviewer #1: Yes

Reviewer #2: Yes

5. Review Comments to the Author

Reviewer #1: The current study has used NEWS2 score while hospital admission of the patients to predict the poor outcome in the COVID19 hospitalization case. They have compared score of 5 (or more) and score of 6 (or more) to predict that score of 5 or more to be more accurate in predicting the poor outcome. Using univariate analysis this study found that around 18 factors associated with the poor outcome, whereas, multiple regression analysis showed that only 5 factors are associated with poor outcome. Authors also claimed the study to be first of its kind in India. There are different studies, which also cited the use of NEWS2 score either 5 (or more) or 6 (or more) to predict poor outcomes. Most of these studies are retrospective in nature, whereas the current study is perspective in nature. Although I found the study interesting, I have few queries that may be explained by the authors.

1. One of the major claim by the study is that a NEWS2 score of 5 or more can predict the poor outcome in the case of COVID19 patients better than the NEWS2 score of 6. There is no comparison between score 5 and 6 mentioned in the result section of this study. If the authors claiming so, need to give a comparison between both the scores showing how one is better than other. “The baseline NEWS2 score was 5 or more in 136 of the 399 (34.1%) patients while it was 6 or more in 115 of the 399 (28.8%) patients.” How many poor outcomes were there in 21 extra patients that was identified by score 5? Can this score be used by the doctors for taking any decision?

2. The result section is poorly mentioned and can be elaborated to make it easily understandable for the readers.

3. Why the odds ratio in the case of ARDs on Chest X-ray is so high, how to explain it? Can the authors describe how they analysed the odds ratio in the result section?

4. There are several studies that suggest that one or more comorbidity condition leads to higher rate of poor outcome. However, this study found that co-morbidities is not associated with the poor outcomes. How authors will explain this? Can they provide any further supporting evidence in this regard?

5. Table 4: Can the authors add one more column to show the %age of each variables in total population. It will be easier for the readers to read and compare the data.

Reviewer #2: The manuscript PNE - E-21-23108 is an attempt to predict poor outcome in hospitallised COVID-19 patients in India using the NEWS-2 scoring system .

This is a good attempt at using the NEWS-2 score in a study over 399 patients with 50% of the patients being diabetics and another 44 % of the patents being hypertensive.

The following observations are made regarding the manuscript :

1. The score of 5 has been used to predict the prognosis in the patients. The authors have also stated that the AUC for a score of 5 was better than 6. How have the authors substantiated this finding ? Details may be sent

2. Other co-morbidities have also been stated in the results and discussion section. Will the inclusion of these as stated for Fig-2 and table -5 change the score to more than 5 as a predictor of poor outcome on THE NEWS-2 scoring system

3. The authors have not included laboratory parameters for the NEWS-2 score. How would the NEWS-2 score compare with another scoring system that combines laboratory with clinical parameters for predicting poor outcome

4. The conclusion is withered and needs to be toned based on the definite results obtainedin the study

6. PLOS authors have the option to publish the peer review history of their article (what does this mean?). If published, this will include your full peer review and any attached files.

Reviewer #1: No

Reviewer #2: No

---

## [Author Response · Author response to Decision Letter 0]

9 Sep 2021

Reviewer 1

Reviewer’s comments 

1. One of the major claim by the study is that a NEWS2 score of 5 or more can predict the poor outcome in the case of COVID19 patients better than the NEWS2 score of 6. There is no comparison between score 5 and 6 mentioned in the result section of this study. If the authors claiming so, need to give a comparison between both the scores showing how one is better than other. “The baseline NEWS2 score was 5 or more in 136 of the 399 (34.1%) patients while it was 6 or more in 115 of the 399 (28.8%) patients.” How many poor outcomes were there in 21 extra patients that was identified by score 5? Can this score be used by the doctors for taking any decision?

Response: We thank the reviewer for this observation. We have now included the comparison of diagnostic accuracy measures of the two cut-offs (5 and 6). This has been added as a table in the manuscript. 

The sensitivity of score 5 or more was 93.3% compared to 90% for score of 6 or more. Since the score is proposed to be used as a screening tool for identifying patients who are likely to have poor outcomes, we chose the score cut off with higher sensitivity. 

This cut off however, identified one poor outcome (death) among the 21 patients. 

We believe that when used in the real life setting where case numbers are very high, this might still be relevant.

2. The result section is poorly mentioned and can be elaborated to make it easily understandable for the readers. 

 Response: We have elaborated as suggested to make it more understandable 

3. Why the odds ratio in the case of ARDs on Chest X-ray is so high, how to explain it? Can the authors describe how they analysed the odds ratio in the result section?

 Response: Of the 399 patients in the study, poor outcomes (death or ventilation) was noted in 30. 

8 of the 30 who had poor outcomes had ARDS; only 3 out of the 369 who had good outcomes had ARDS. 

This considerable difference is the possible reason for the very high Odds Ratio. 

We have described this in results section.

4. There are several studies that suggest that one or more comorbidity condition leads to higher rate of poor outcome. However, this study found that co-morbidities is not associated with the poor outcomes. How authors will explain this? Can they provide any further supporting evidence in this regard?

Response: We agree that several studies have identified such association with co-morbidities. 

The absence of association in our case could be due to the outcomes chosen. We have included both mechanical ventilation and death as poor outcomes since we thought these are important outcomes in hospitalised patients. 

Different studies have used various outcomes (such as severe disease or death alone) explaining the diversity in results. 

The other explanation is the high prevalence of co-morbidities in the general population in this area. Many studies have shown that diabetes and hypertension are highly prevalent in general population in India, especially South India. 

In this study, overall around 50% of patients were diabetic and 44% hypertensive. Therefore it was unable to detect any significant difference between those with and without poor outcomes. 

5. Table 4: Can the authors add one more column to show the %age of each variables in total population. It will be easier for the readers to read and compare the data. 

Response: This column has been added and table is now modified as table 5; however since there are few continuous variables, these could not be represented in percentages. 

Reviewer 2

Reviewer’s comments 

1. The score of 5 has been used to predict the prognosis in the patients. The authors have also stated that the AUC for a score of 5 was better than 6. How have the authors substantiated this finding ? Details may be sent 

Response: We thank the reviewer for this observation. 

We have now included the comparison of diagnostic accuracy measures of the two cut-offs (5 and 6). This has been added as a table (Table 4) in the manuscript. 

The sensitivity of score 5 or more was 93.3% compared to 90% for score of 6 or more. Since the score is proposed to be used as a screening tool for identifying patients who are likely to have poor outcomes, we chose the score cut off with higher sensitivity. 

2. Other co-morbidities have also been stated in the results and discussion section. Will the inclusion of these as stated for Fig-2 and table -5 change the score to more than 5 as a predictor of poor outcome on THE NEWS-2 scoring system

Response: The standard NEWS2 score is validated only for the clinical variables included and does not include any of the co-morbidities like hypertension, diabetes etc. therefore we could not include them into the score. 

Further, studies have shown that adding pre-existing co-morbidities to the model made no difference to risk prediction. This has been mentioned in discussion section. 

3. The authors have not included laboratory parameters for the NEWS-2 score. How would the NEWS-2 score compare with another scoring system that combines laboratory with clinical parameters for predicting poor outcome

Response: The NEWS2 score is a clinical bedside scoring that includes only the respiratory rate, oxygen saturation (SpO2), need for supplemental oxygen, pulse rate, level of consciousness and temperature. 

Besides the objective of our study was to determine the ability of NEWS2 score to predict poor outcomes and therefore we did not compare it with any other scores that use laboratory parameters. The advantage of NEWS2 score is that since it does not include laboratory parameters, it can be used readily in resource poor and emergency settings as well. 

4. The conclusion is withered and needs to be toned based on the definite results obtained in the study

Response: Thank you for the observation. We have modified the conclusion in alignment with the results of the study.

---

## [Decision Letter · Decision Letter 1]

20 Oct 2021

PONE-D-21-23108R1National Early Warning Score 2 (NEWS2) to predict poor outcome in hospitalised COVID-19 patients in IndiaPLOS ONE

Dear Dr. Basheer,

Thank you for submitting your manuscript to PLOS ONE. After careful consideration, we feel that it has merit but does not fully meet PLOS ONE’s publication criteria as it currently stands. Therefore, we invite you to submit a revised version of the manuscript that addresses the points raised during the review process.

Please revise.

We look forward to receiving your revised manuscript.

Kind regards,

Academic Editor

PLOS ONE

Journal Requirements:

Reviewers' comments:

Reviewer's Responses to Questions

**Comments to the Author**

1. If the authors have adequately addressed your comments raised in a previous round of review and you feel that this manuscript is now acceptable for publication, you may indicate that here to bypass the “Comments to the Author” section, enter your conflict of interest statement in the “Confidential to Editor” section, and submit your "Accept" recommendation.

Reviewer #1: (No Response)

Reviewer #2: All comments have been addressed

2. Is the manuscript technically sound, and do the data support the conclusions?

Reviewer #1: Partly

Reviewer #2: Yes

3. Has the statistical analysis been performed appropriately and rigorously? 

Reviewer #1: N/A

Reviewer #2: Yes

4. Have the authors made all data underlying the findings in their manuscript fully available?

Reviewer #1: Yes

Reviewer #2: Yes

5. Is the manuscript presented in an intelligible fashion and written in standard English?

Reviewer #1: Yes

Reviewer #2: Yes

6. Review Comments to the Author

Reviewer #1: In the revised version still the result section is not elaborated. Thus it is difficult to understand. How the sensitivity and specificity were determined? Can the authors describe the calculations in the method section? How the sensitivity and specificity was calculated for the poor outcome? What are their significance? Can authors describe this in the result section? Line 160-168 needs more clarifications.

How the Odd ratio was calculated (line 192) is not clear. Can the authors describe it in the method section or in the result section?

Legend for both the figures should be mentioned to understand the figures.

The explanation for Figure 2 is not clear. What the AUC is mentioning here? Is it for score 5 or 6?

Line 231: been a tough ask to predict: change ask to task

As the current study did not find any association of the comorbidity with the poor outcomes, a statement should be made in the discussion section.

The conclusion can be more elaborative, should present numbers. This will help the readers to understand the concluding remarks of the article.

Reviewer #2: The previous comments from the reviewer have been addressed fully and completely. The manuscript has been modified to read the same

7. PLOS authors have the option to publish the peer review history of their article (what does this mean?). If published, this will include your full peer review and any attached files.

Reviewer #1: No

Reviewer #2: No

---

## [Author Response · Author response to Decision Letter 1]

10 Nov 2021

1. In the revised version still the result section is not elaborated. Thus it is difficult to understand

We regret that the previous revision failed to improve understanding. 

We have added more details on number of patients who had score of 5 and above and went on to develop poor outcomes. Similar description has been given for score of 6 and above. We hope these will improve the understanding of the section. 

Please see:

Lines 166 and 167

Lines 176 to 180

Lines 183 to 185

2. How the sensitivity and specificity were determined? Can the authors describe the calculations in the method section? How the sensitivity and specificity was calculated for the poor outcome? What are their significance? Can authors describe this in the result section? 

Sensitivity refers to a test's ability to designate an individual with disease as positive. In this study it meant the ability of NEWS2 score to identify the proportion of people with the disease who will have a poor outcome. A highly sensitive test means that there are few false negative results, and thus fewer cases of disease are missed and is good at including most people who have the condition. 

In this study, in order to calculate the sensitivity of a score of 5, we determined the proportion of patients with a baseline score of 5 and above who developed poor outcome. Poor outcome was defined as death or mechanical ventilation (mentioned in methods). Similarly, sensitivity was calculated for score of 6.

On the other hand, Specificity refers the ability of a test to correctly identify people without the disease. In this study it meant the ability of NEWS2 score to identify the proportion of people with the disease who will have a better outcome. 

Hence in this study, in order to find the specificity of score of 5, we determined the proportion of patients with a baseline score of 5 and above who did not develop a poor outcome (or had good outcome). Same method was used for specificity of score 6. 

We have added this explanation in the methods section and also elaborated the calculation in results and added supplementary tables (S2 tables). 

Please see:

Lines 183 and 184.

3. Line 160-168 needs more clarifications

In this study Table 4 and lines 160 – 166 explains about best cut off for the NEWS2 score.

We have now added descriptions of how the calculations were made. 

Please see:

Lines 176 to 180

Lines 184 and 185 

Supplementary tables 1 and 2 

4. How the Odd ratio was calculated (line 192) is not clear. Can the authors describe it in the method section or in the result section?

The odds ratio quantifies the relationship between an exposure and a disease and tells us how much higher the odds of exposure are among cases than among controls. In this study those who need for mechanical ventilation or death within 28 days (i.e., Poor outcome) are considered as the case-patients and patients discharged (i.e., Good outcome) are considered as the control-patients. Exposure status is defined as findings of the Chest X-ray viz., ARDS and no ARDS. Logistic regression, a standard technique to calculate odds ratio, was employed to quantify the relationship between Chest X-ray and poor outcome and expressed as Odds ratio with 95% Confidence intervals and p values.

This information has been added in methods section covering Odds ratio calculation for various variables.

Please see:

Lines 122 to 130

5. Legend for both the figures should be mentioned to understand the figures.

The explanation for Figure 2 is not clear. What the AUC is mentioning here? Is it for score 5 or 6?

Legend for figure 1 has been elaborated. 

Lines 105 and 106

Regarding Legend for figure 2:

The Area Under the Curve (AUC) is the area under the receiver operating characteristic curve (ROC curve). It is a number between zero and one, because the ROC curve fits inside a unit square. In general, an AUC of 0.5 suggests no discrimination (i.e., ability to diagnose patients with and without the disease or condition based on the test), 0.7 to 0.8 is considered acceptable, 0.8 to 0.9 is considered excellent, and more than 0.9 is considered outstanding. Thus higher the AUC, the better the performance of the model at distinguishing between the positive and negative cases. In this study the AUC to predict poor outcome i.e., need for mechanical ventilation or death within 28 daysbased on NEWS2 score at admission was 0.887 which is considered excellent.

The sensitivity and specificity can be calculated for each cut off value of the test score and it does not depend upon AUC. In other words, AUC is not for a specific cut off score. The typo error “for this cut off” has therefore been removed since this is not applicable for AUC. 

We regret the confusion caused by the previous wording.

Please see:

Line 187 and 188.

6. Line 231: been a tough ask to predict: change ask to task

We thank the reviewer for identifying this mistake. We have now changed it. 

Please see:

Line 273

7. As the current study did not find any association of the comorbidity with the poor outcomes, a statement should be made in the discussion section.

We thank you for this suggestion. 

In response to a similar suggestion in the first review, we have already added this in the discussion. 

Please see:

Line 338 to 343.

8. The conclusion can be more elaborative, should present numbers. This will help the readers to understand the concluding remarks of the article

We thank you for the valuable suggestion. We have now added the numbers pertaining to sensitivity and specificity which are the primary outcomes of the study. We have not included more details as it may be a repetition of results. 

Please see: 

Line 354 to 356

---

## [Decision Letter · Decision Letter 2]

1 Dec 2021

National Early Warning Score 2 (NEWS2) to predict poor outcome in hospitalised COVID-19 patients in India

PONE-D-21-23108R2

Dear Dr. Basheer,

We’re pleased to inform you that your manuscript has been judged scientifically suitable for publication and will be formally accepted for publication once it meets all outstanding technical requirements.

Kind regards,

Academic Editor

PLOS ONE

Additional Editor Comments (optional):

Reviewers' comments:

Reviewer's Responses to Questions

**Comments to the Author**

1. If the authors have adequately addressed your comments raised in a previous round of review and you feel that this manuscript is now acceptable for publication, you may indicate that here to bypass the “Comments to the Author” section, enter your conflict of interest statement in the “Confidential to Editor” section, and submit your "Accept" recommendation.

Reviewer #1: All comments have been addressed

Reviewer #3: All comments have been addressed

2. Is the manuscript technically sound, and do the data support the conclusions?

Reviewer #1: Yes

Reviewer #3: Yes

3. Has the statistical analysis been performed appropriately and rigorously? 

Reviewer #1: Yes

Reviewer #3: Yes

4. Have the authors made all data underlying the findings in their manuscript fully available?

Reviewer #1: Yes

Reviewer #3: Yes

5. Is the manuscript presented in an intelligible fashion and written in standard English?

Reviewer #1: Yes

Reviewer #3: Yes

6. Review Comments to the Author

Reviewer #1: In the revised manuscript the authors have addressed all the queries raised during previous revision.

Reviewer #3: The large sample size play a major determinant for your study to be accepted.

Relating COVID cases with simple, doable scores for every centres is important especially with high patients load.

My suggestion is to only include significant and non significant but important factors for table 5 & 6

7. PLOS authors have the option to publish the peer review history of their article (what does this mean?). If published, this will include your full peer review and any attached files.

Reviewer #1: No

Reviewer #3: No

---

## [Editor Report · Acceptance letter]

3 Dec 2021

PONE-D-21-23108R2 

National Early Warning Score 2 (NEWS2) to predict poor outcome in hospitalised COVID-19 patients in India 

Dear Dr. Basheer:

I'm pleased to inform you that your manuscript has been deemed suitable for publication in PLOS ONE. Congratulations! Your manuscript is now with our production department. 

Kind regards, 

on behalf of

Dr. Robert Jeenchen Chen 

Academic Editor

PLOS ONE